# DEEP FAIR DISCRIMINATIVE CLUSTERING

## ABSTRACT

Deep clustering has the potential to learn a strong representation and hence better clustering performance than traditional clustering methods such as $k$-means and spectral clustering. However, this strong representation learning ability may make the clustering unfair by discovering surrogates for protected information which our experiments empirically show. This work studies a general notion of group-level fairness for both binary and multi-state protected status variables (PSVs). We begin by formulating the group-level fairness problem as an integer linear programming whose totally unimodular constraint matrix means it can be efficiently solved via linear programming. We then show how to inject this solver into a discriminative deep clustering backbone and hence propose a refinement learning algorithm to combine the clustering goal with the fairness objective to learn fair clusters adaptively. Experimental results on real-world datasets demonstrate that our model consistently outperforms state-of-the-art fair clustering algorithms. Furthermore, our framework shows promising results for novel fair clustering tasks including flexible fairness constraints, multi-state PSVs and predictive clustering.

## 1 INTRODUCTION

Clustering is essential as it is the basis of many AI tools and has been widely used in real-world applications involving humans (Jain et al., 1999) such as market research, social network analysis, and crime analysis. However, as AI tools augment and even replace humans in decision-making, the need to ensure clustering is fair becomes paramount. Here fairness is measured using protected status variables (PSVs) such as gender, race, or education level. Fairness takes two primary forms (Binns, 2020): i) group-level fairness and ii) individual-level fairness. In this paper, we study the former which ensures that no one cluster contains a disproportionately small number of individuals with protected status. Motivated by this goal, our work aims to add fairness rules to deep clustering.

Recent works (Rösner & Schmidt, 2018; Schmidt et al., 2019; Kleindessner et al., 2019b; Backurs et al., 2019; Bera et al., 2019) have been proposed for non-deep fair clustering algorithms. To ensure group-level fairness, many of these works use the notion of the *disparate impact doctrine* encoded as a constraint, that instances from different protected groups must have approximately (within a tolerance) equal representation in a cluster compared to the population. Different geographic regions place this tolerance at different levels (Chierichetti et al., 2017). These existing algorithms optimize the clustering quality by minimizing some well-known clustering objectives while satisfying the group-level fairness constraints. Previous examples of adding fairness to clustering algorithms include k-median based approaches (Chierichetti et al., 2017; Backurs et al., 2019; Bera et al., 2019) and spectral clustering based algorithm (Kleindessner et al., 2019b). However, all these works evaluate their performance on low-dimensional tabular data and (Chierichetti et al., 2017; Kleindessner et al., 2019b; Backurs et al., 2019) study the problems only with binary PSV.

Deep clustering (Xie et al., 2016; Hu et al., 2017; Guo et al., 2017; Wang et al., 2019) has the ability to simultaneously cluster and learn a representation for problems with large amounts of complex data (i.e., images, texts, graphs). However, the representation learning ability sometimes makes the learner suffer from bias hidden in the data which can lead to unfair clustering results. For example, clustering of portraits may create clusters based on features which are surrogates for racial and other protected status information. One way to overcome this is by adding group-level fairness to deep clustering which is a challenging and understudied problem. A significant challenge is it is hard to translate the current fair clustering algorithms into an end-to-end deep clustering setting. For example, geometric pre-processing steps such as computing fairlets (Chierichetti et al., 2017) to ensure fairness will not

work as the end-to-end learning of deep learners means the underlying features that clustering is performed on are unknown apriori. Similarly, another line of work that adds constraints into deep learning models such as (Xu et al., 2018; Zhang et al., 2019) are not appropriate either as these constraints are at the *instance* level, whereas we require to apply fairness rules at a cluster level.

The work on fair deep clustering is relatively new. The first work on fair deep clustering (Wang & Davidson, 2019) studies deep fair clustering problem from a geometric perspective which aims to learn a fair representation with multi-state PSV. The most recent work (Li et al., 2020) proposes a deep fair visual clustering model with adversarial learning to encourage the clustering partition to be statistically independent of each sensitive attribute (PSV). Although these deep clustering approaches demonstrate better clustering performance compared to the traditional fair clustering algorithms (Table 1), their fairness results are relatively poor compared to those fair clusterings with fairness guarantees (Chierichetti et al., 2017; Backurs et al., 2019). Our work can be seen as combining the benefits of deep learning and discrete optimization to produce *guaranteed fair predictions* on clustered data with PSVs while making *out-of-sample fair predictions* for data without PSVs.

In this paper, we propose a novel deep fair clustering framework to address the above issues. We adopt a probabilistic discriminative clustering network and learn a representation that naturally yields compact clusters. To incorporate the group-level fairness rules in the deep learner, we first formulate our fairness objective as an integer linear programming (ILP) problem that guarantees group-level fairness. This ILP is efficient to solve as its constraint matrix is totally unimodular. Further, we propose a refinement learning algorithm to combine the solved fair assignments and the clustering objective. Experimental results on real-world datasets demonstrate that our model achieves guaranteed fair results with competitive clustering performance. We also examine the novel uses of our framework in predictive clustering, flexible fair clustering, and challenging tasks with multi-state PSVs. The major contributions of this paper are summarized as follows:

- We optimize a general notion of fairness for multi-state PSVs which we prove is equivalent to optimize the general balance measure (Chierichetti et al., 2017) for *disparate impact*.

- We formulate our fairness assignment sub-problem as an ILP which is NP-hard. We show that by relaxing our constraint matrix to be *totally unimodular* our sub-problem can be efficiently solved by an LP solver (but still generate integer solutions). (See Theorem 4.2).

- We propose an end-to-end refinement learning algorithm that combines deep learning and discrete optimization to learn fair representation for clustering. (See Algorithm 1).

- Extensive experimental results show that our work can achieve guaranteed fairness with competitive clustering performance. We demonstrate our novel extensions for fair clustering tasks in predictive clustering, multi-state PSVs and flexible fairness rules. (See Section 5.2).

In the next section 2 we discuss the related work. Then we outline our measure of fairness and how it relates to classic measures of disparate impact in section 3. In our approach section 4, we introduce our clustering framework and encode our fairness objective as an ILP which can be efficiently solved via our relaxation. A refinement learning algorithm is proposed for end-to-end fair clustering. Finally we empirically evaluate the effectiveness of our approach in section 5 and conclude in section 6.

## 2 RELATED WORK

Fair clustering has received much attention recently (Schmidt et al., 2019; Kleindessner et al., 2019a; Ahmadian et al., 2019; Chen et al., 2019; Davidson & Ravi, 2020; Mahabadi & Vakilian, 2020; Brubach et al., 2020). Chierichetti et al. (2017) first addressed the disparate impact for clustering problems in the presence of binary PSVs. Their work apriori groups instances into many fairlets which are used as input into standard k-medians style algorithms. Their work is guaranteed to produce a specified level of fairness and achieve a constant factor approximation with respect to cluster quality. Backurs et al. (2019) improves the fair decomposition algorithm to linear run-time. Later on, Bera et al. (2019) propose a general fair clustering algorithm that allows human-specified upper and lower bounds on any protected group in any cluster. Their work can be applied to any clustering problems under $\ell_p$ norms such as k-median, k-means, and k-center. Besides the centroid-based method, Kleindessner et al. (2019b) extends the fairness notion to graph spectral clustering problems. Ziko et al. (2021) propose a general, variational and bound-optimization framework of fair clustering.

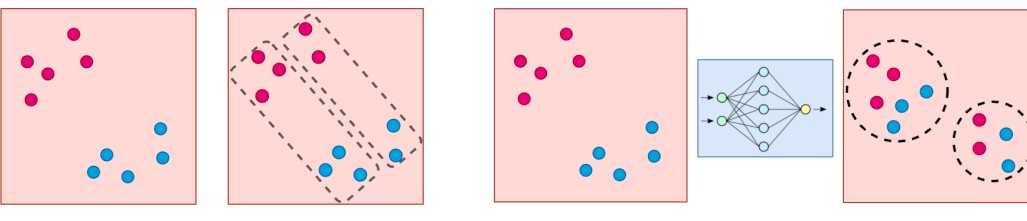

(a) Traditional Fair Clustering        (b) Deep Fair Clustering

Figure 1: Note the red and blue points are instances with different PSV values. Traditional fair clustering (left) aims to find a fair partition of the data while minimizing some classic clustering objectives. Deep fair clustering (right) aims to learn a general fair representation to cluster the data.

Previous fair clustering approaches mainly focus on adding fairness constraints into traditional clustering algorithms. In our work, we aim to study the fairness problem for recently proposed deep clustering algorithms (Xie et al., 2016; Yang et al., 2017; Hu et al., 2017; Caron et al., 2018; Shaham et al., 2018; Tzoreff et al., 2018; Shah & Koltun, 2018). Deep clustering algorithms connect representation learning and clustering together and have demonstrated their advantages over the two-phase clustering algorithms which use feature transformation first and then clustering. The goal of deep fair clustering is to learn a fair and clustering-favored representation. We illustrate the basic intuitions behind traditional fair clustering methods and deep fair clustering approaches in Figure 1.

One of the earliest works (Wang & Davidson, 2019) to address the deep fair clustering problem learns a latent representation such that the cluster centroids are equidistant from every "fairoid" (the centroid of all the data belonging to the same protected group). Recently, Li et al. (2020) encodes the fairness constraints as an adversarial loss and concatenates the fairness loss to a centroid-based deep clustering objective as a unified model. Unlike previous deep fair clustering works, we translate the fairness requirements into an ILP problem that generates guaranteed fair solutions given the PSVs. Meanwhile, our formulation allows for a general notion of fairness that supports flexible fairness constraints and multi-state PSVs. Moreover, we propose a novel learning framework to train fair clustering models via simultaneous clustering and fitting the self-generated fairness signals.

## 3 DEFINITIONS OF GROUP-LEVEL FAIRNESS

We begin this section by overviewing the seminal definition of group-level fairness in clustering (see equation 1) and then its extension to multi-state PSVs (see equation 2). We then go onto show a new measure that our deep clustering framework will optimize (see equation 3) and equation 2 have the same optimal condition as shown in Theorem 3.2.

### 3.1 NOTION OF FAIRNESS

Let $X \in \mathbb{R}^{N \times D}$ denote $N$ data points with $D$ dimension features. The prediction function $\phi$ assigns each instance to one unique cluster, $\phi : x \to \{1, ...K\}$, which forms $K$ disjoint clusters $\{C_1, ...C_K\}$. Given the protected status variable (denoted as PSV) $\mathcal{A}$ with $T$ states, $X$ can be partitioned into $T$ demographic groups as $\{G_1, G_2, ...G_T\}$.

**Definition 1.** *The seminal proposed measure of fairness for clustering with binary PSV (Chierichetti et al., 2017) encoded disparate impact as follows:*

$$balance(C_k) = \min(\frac{N_k^1}{N_k^2}, \frac{N_k^2}{N_k^1}) \in [0, 1] \tag{1}$$

Here $N_k^1$ and $N_k^2$ represent the populations of the first and second demographic groups in cluster $C_k$. Such a measure of fairness only works for binary PSV. To allow for multi-state PSVs, let $N_k^{min} = \min(N_k^1 \ldots N_k^T)$ denotes the smallest (in size) protected group in cluster $k$ and $N_k^{max} = \max(N_k^1 \ldots N_k^T)$ denotes the largest group. We extend the balance measure for multi-state PSV as:

$$balance(C_k) = \frac{N_k^{min}}{N_k^{max}} \in [0, 1] \tag{2}$$

Recent works (Rösner & Schmidt, 2018; Bera et al., 2019) also propose a new fairness measure to allow for fair clustering problems with multi-state PSVs.

**Definition 2.** *Let $\rho_i$ be the representation of group $G_i$ in the dataset as $\rho_i = |G_i|/N$, and $\rho_i(k)$ be the representation of group $G_i$ in the cluster $C_k$: $\rho_i(k) = |C_k \cap G_i|/|C_k|$. Using these two values, the fairness value for cluster $C_k$ is:*

$$fairness(C_k) = \min(\frac{\rho_i}{\rho_i(k)}, \frac{\rho_i(k)}{\rho_i}) \in [0, 1] \quad \forall i \in \{1, \dots T\} \tag{3}$$

The overall fairness of a clustering is defined as the *minimum* fairness value over all the clusters. Similarly, the overall balance is the *minimum* balance value of all the clusters.

### 3.2 EQUIVALENCE OF OPTIMIZING FAIRNESS AND BALANCE MEASURES

Here we show that optimizing equation 3 is equivalent to optimizing our extended definition of balance in equation 2. We see that equation 3 achieves maximal fairness when $P(x \in G_t | x \in C_k) = \rho_t$. Our balance measure in equation 2 achieves optimal balance when $P(x \in G_t | x \in C_k) = \frac{1}{T}$ for any protected group $G_t$ in cluster $C_k$. However, this is an ideal case as protected groups may be imbalanced. Denote the size of each protected group as $|G_i|$ and the size of the data set as $N$, we now show that the optimal balance is achieved if and only if $P(x \in G_t | x \in C_k) = \rho_t$. This result indicates the equivalence of optimizing fairness (equation 3) and generalized balance (equation 2).

**Lemma 3.1.** *The optimal balance can be achieved only when all the clusters have the same balance. Formally, $\forall i, j \in \{1, 2, ..., K\}$: $balance(C_i) = balance(C_j)$.*

**Theorem 3.2.** *To achieve optimal balance value for multi-state protected variables, we must satisfy the condition: $P(x \in G_t | x \in C_k) = \rho_t$ which is precisely the optimal fairness value for equation 2.*

## 4 DEEP FAIR CLUSTERING ALGORITHM

We introduce our framework in this section. Our approach can be viewed as learning fair clustering under a discriminative clustering loss objective and a fairness objective with self-generated signals.

### 4.1 REVIEW OF BASE CLUSTERING MODEL

For base clustering model, we directly apply the previous work (Hu et al., 2017) and overview it here. We learn a neural network $f_\theta$ as a discriminative function to predict the clustering assignments $Y = \sigma(f_\theta(X)) \in \mathbb{R}^{N \times K}$ based on input $X \in \mathbb{R}^{N \times D}$ and softmax function $\sigma$. The mutual information $I(X; Y)$ between $X$ and $Y$ is calculated as the difference between marginal entropy $H(Y)$ and conditional entropy $H(Y|X)$:

$$I(X; Y) = H(Y) - H(Y|X) = h(\frac{1}{N} \sum_{i=1}^{N} \sigma(f_\theta(x_i))) - \frac{1}{N} \sum_{i=1}^{N} h(\sigma(f_\theta(x_i))) \tag{4}$$

where $h$ is the entropy function. With weight decay term the clustering objective $\ell_C$ is as follows:

$$\ell_C = \frac{1}{N} \sum_{i=1}^{N} h(\sigma(f_\theta(x_i))) - h(\frac{1}{N} \sum_{i=1}^{N} \sigma(f_\theta(x_i))) + \alpha \sum_{l=1}^{L} \|\theta^l\|^2 \tag{5}$$

where $\alpha$ denotes the hyper-parameter for network parameters $\{\theta^1 \dots \theta^L\}$. Maximizing $H(Y)$ will punish imbalanced cluster size and prevent trivial solutions where all the instances are clustered into one cluster while minimizing $H(Y|X)$ will map similar instances $x$ to have similar labels $y$.

Further, self-augmented training is applied to encourage the representations to be locally invariant. Here we add a local perturbation of instance $x$ such that $x' = x + t$ and hope to maximize the perturbation $t$ subject to the constraint that the clustering assignments for $x$ and $x'$ are the same. Virtual adversarial training (Miyato et al., 2018) is applied to generate adversarial direction for $t$. Denote the current model's parameters $\theta$ to help estimate the true clustering indicator vector for instance $x$ as $\sigma(f_\theta(x))$, the formulation to compute the adversarial perturbation $t_{adv}$ is as follows:

$$t_{adv} = \underset{t; ||t||_2 \leq \epsilon}{\arg\max} \ \text{KL}(\sigma(f_\theta(x)), \sigma(f_\theta(x + t))) \tag{6}$$

With the generated $t_{adv}$, we have the augmentation loss $\ell_{Aug}$ which minimizes the KL divergence between clustering assignment $\sigma(f_\theta(x_i))$ and its augmented version's assignment $\sigma(f_\theta(x_i'))$:

$$\ell_{Aug} = \sum_{i=1}^{N} \text{KL}(\sigma(f_\theta(x_i)), \sigma(f_\theta(x_i'))) \tag{7}$$

Finally, the base clustering model optimizes the clustering loss $\ell_C$ and $\ell_{Aug}$ simultaneously. Note that we favor this probabilistic discriminative clustering model (Hu et al., 2017) since it has fewer assumptions about the natures of categories that are made and fits our fairness objective which requires fractional clustering assignments as inputs to indicate the degree of cluster assignment belief.

## 4.2  GENERATING FAIR ASSIGNMENTS UNDER GROUP-LEVEL FAIRNESS CONSTRAINTS

Let the fractional clustering assignments from the current learned model be $Y = \{y_1, \ldots y_N\} \in \mathbb{R}^{N \times K}$. To use these to form fair clustering assignments, we solve a fairer assignment matrix $\hat{Y} = \{\hat{y}_1, \ldots \hat{y}_N\} \in \mathbb{Z}^{N \times K}$ that satisfy the fair optimal condition: $P(x \in G_t | x \in C_k) = \rho_t$. To address the fair assignment problem we formulate it as an integer linear programming where we aim to minimize the changes to the current assignment $Y$ to obtain a fairer assignment $\hat{Y}$ as follows:

$$\text{Objective:} \quad \underset{\hat{Y}}{\arg\min} \sum_{i=1}^{N} [1 - y_i \times \hat{y_i}^T] \tag{8}$$

Recall $y_i$ is a row vector which represents the probability distribution over the cluster assignments for instance $i$ and $\hat{y}_i$ chooses exactly one cluster to assign instance $i$ to. Naturally the objective is maximized when $y_i$ is assigned to its most probable cluster but this may cause an unfair clustering.

We denote $\rho_i = |G_i|/N$ as the fraction of the protected group $G_i$ in the data set and our aim is for each cluster to have the same density. Let $M \in \mathbb{Z}^{N \times T}$ encode the sensitive attributes for the entire population such that $M_{it} \in \{0, 1\}$ indicates whether an instance $x_i$ belongs to a protected group $G_t$. To satisfy optimal fair condition $P(x \in G_t | x \in C_k) = \rho_t$ we have the following constraints:

$$\sum_{i=1}^{N} M_{it}\hat{y}_{ij} = \sum_{i=1}^{N} \hat{y}_{ij}\rho_t \quad \forall j \in \{1 \ldots K\}, t \in \{1 \ldots T\} \tag{9}$$

Now we relax the problem by fixing the size of each new cluster to make the constraint matrix totally unimodular. We round the soft probabilistic assignment $Y$ as hard assignments $Y'$ by assigning the cluster with largest probability. Then the size of cluster $C_j$ is $|C_j| = \sum_{i=1}^{N} y'_{ij}$. The constraints for new clusters' size are:

$$\sum_{i=1}^{N} \hat{y}_{ij} = |C_j| \quad \forall j \in \{1 \ldots K\} \tag{10}$$

Lastly we add constraints for $\hat{Y}$ to ensure each instance is assigned to one cluster:

$$\sum_{j=1}^{K} \hat{y}_{ij} = 1 \quad \forall i \in \{1 \ldots N\} \tag{11}$$

Note this ILP formulation also supports user-defined $\rho_t$ which can be seen as a flexible fairness rule. Next we show the constraint matrix of our ILP problem is totally unimodular so that we can efficiently solve it with a LP solver and still return integral solutions.

We know that if a constraint matrix of an ILP is totally unimodular (TU) then we can solve the problem using an LP (linear program) solver and the solution will still be integral (Schrijver, 1998). Using an LP solver will largely reduce the running time and (Vaidya, 1989) has shown that the running time for LP is polynomial in the input size. In the above proposed constraints, there are $NK$ unique regular variables ($N$ instances and $K$ categories). To construct the constraint matrix $C$ which encodes constraint 9, 10 and 11, we will use $NK$ regular variables. Matrix $C$ has $T + 1$ rows (the first $T$ rows correspond to the fairness constraints in equation 9 and last row corresponds to

---

**Algorithm 1** Main learning algorithm for deep fair discriminative clustering.

---

**Input:** Input $\{x_k\}_{k=1}^N$, sensitive attributes $M$, cluster size $K$, network structure $f$, hyper-parameters $\alpha, \beta, \gamma$.
**Output:** Clustering network $f_\theta$, predictions $\{y_k\}_{k=1}^N$ .
 1: **for** each pre-trained epoch **do**
 2:     **for** sampled mini-batch $\{x_k\}_{k=1}^n$ **do**
 3:         Calculate $\ell_C = \frac{1}{n}\sum_{i=1}^n h(\sigma(f_\theta(x_i))) - h(\frac{1}{n}\sum_{i=1}^n \sigma(f_\theta(x_i))) + \alpha\sum_{l=1}^L \|\theta^l\|^2$
 4:         Generate $x_k' = x_k + t$ via solving $t$ from eq 6.
 5:         Calculate $\ell_{Aug} = \sum_{i=1}^n \mathrm{KL}(\sigma(f_\theta(x_i)), \sigma(f_\theta(x_i')))$
 6:         Update network $f_\theta$ via minimizing $\ell_C + \gamma\ell_{Aug}$.
 7:     **end for**
 8: **end for**
 9: **repeat**
10:     Generate predictions $\{y_k\}_{k=1}^N$ based on $f_\theta$.
11:     Construct a fair assignment problem via objective 8 and constraints defined in eq 9, 10 and 11.
12:     Solve fair assignments $\{\hat{y}_k\}_{k=1}^N$ via LP solver.
13:     **for** sampled mini-batch $\{x_k\}_{k=1}^n$ **do**
14:         Calculate $\ell_{Fair} = \frac{1}{n}\sum_{i=1}^n \hat{y}_i log(\sigma(f_\theta(x_i)))$
15:         Calculate $\ell_C = \frac{1}{n}\sum_{i=1}^n h(\sigma(f_\theta(x_i))) - h(\frac{1}{n}\sum_{i=1}^n \sigma(f_\theta(x_i))) + \alpha\sum_{l=1}^L \|\theta^l\|^2$
16:         Generate $x_k' = x_k + t$ via solving $t$ from eq 6.
17:         Calculate $\ell_{Aug} = \sum_{i=1}^n \mathrm{KL}(\sigma(f_\theta(x_i)), \sigma(f_\theta(x_i')))$
18:         Calculate $\ell = \ell_C + \beta\ell_{Fair} + \gamma\ell_{Aug}$
19:         Update network $f_\theta$ via minimizing $\ell$.
20:     **end for**
21: **until** $\{y_k\}_{k=1}^N$ satisfy *optimal fairness* rules

---

constraints in equation 11) and $N + K$ columns. Note the first $K$ columns of the last row are set to $0$ and the last $N$ columns of first $T$ rows are set to $0$. In matrix $C$, each entry of $C$ is from $\{-1, 0, 1\}$. Moreover, each column only has one non-zero element. This is because: (1) for constraints set in equation 9, each instance only belongs to one protected group, (2) for constraints set in equation 11, there is only one row vector with $K$ elements as $1$ to ensure the valid assignment.

**Lemma 4.1.** *TU Identity (Schrijver, 1998). Let $C$ be a matrix such that all its entries are from $\{0, 1, -1\}$. Then $C$ is totally unimodular, i.e., each square submatrix of $C$ has determinant $0$, $1$, or $-1$ if every subset of rows of $C$ can be split into two parts $A$ and $B$ so that the sum of the rows in $A$ minus the sum of the rows in $B$ produces a vector all of whose entries are from $\{0, 1, -1\}$.*

**Theorem 4.2.** *The matrix $C$ formed by the coefficients of the constraints used to encode our proposed constraints from equation 9, 10 and equation 11 is totally unimodular.*

### 4.3 LEARNING TO BE FAIRER

To learn a fair clustering model we aim to exploit the fairness assignments $\hat{Y}$ to reshape the features learned via clustering networks $f_\theta$. We treat $\hat{Y}$ as "pseudo-labels" to optimize the following cross entropy loss $\ell_{Fair}$ for fairer results:

$$\ell_{Fair} = \frac{1}{N}\sum_{i=1}^N \sum_{j=1}^K \hat{y}_{ij} log y_{ij} = \frac{1}{N}\sum_{i=1}^N \hat{y}_i log(\sigma(f_\theta(x_i))) \tag{12}$$

Simply optimizing the fairness loss $\ell_{Fair}$ will dramatically change the current clustering representations to fit an approximated fair assignment $\hat{Y}$ which harms the clustering properties. Instead, we propose to learn a fairer and clustering-friendly representation simultaneously by combining the clustering loss $\ell_C$, augmentation loss $\ell_{Aug}$ and fairness loss $\ell_{Fair}$. Note the fair assignments $\hat{Y}$ are updated after each training epoch as the "nearest" fair assignments for current clustering predictions.

To train our proposed framework, we start with the training on base clustering network $f(\theta)$ via optimizing the clustering loss $\ell_C$ and augmentation loss $\ell_{Aug}$ to ensure the data are separated into different meaningful clusters; as clustering model converges we generate fair assignments after each training epoch based on the objective in equation 8 and optimize the overall loss function $\ell$ by concatenating the fairness loss $\ell_{Fair}$ to clustering objectives as $\ell = \ell_C + \beta\ell_{Fair} + \gamma\ell_{Aug}$ where $\beta, \gamma$ are positive weight hyper-parameters. Algorithm 1 summarizes the proposed learning method.

## 5 EXPERIMENTS

We conduct experiments[1] to evaluate our approach empirically and report the following key results:

- Our proposed approach achieves better clustering performance and guaranteed fairness results compared against both traditional fair clustering and deep fair clustering baselines.

- Our proposed approach is effective in novel fair clustering settings such as supporting flexible fairness constraints, clustering with multi-state PSVs, and predictive clustering.

- We show how our learned embedding converges to a latent space useful for fair clustering quickly and also provides insights on tuning hyper-parameter $\beta$ in unsupervised way to achieve our fairness goal with a minimum loss on clustering performance.

### 5.1 EXPERIMENTAL SETUP

We first evaluate our work on two visual data sets with binary PSV that has been used in recent deep fair clustering work (Li et al., 2020): 1) MNIST-USPS consists of 67291 training images of hand-written digits. We use the image source (MNIST or USPS) as a binary PSV and cluster the data into 10 classes representing 10 digits. 2) Reverse-MNIST takes the 60000 training images from MNIST and creates an inverted duplicate to build this dataset. The binary PSV is then original or inverted and the total number of classes is 10. Moreover, we evaluate one challenging fair clustering task with multi-state PSV on the HAR dataset used in (Wang & Davidson, 2019): 3) HAR contains 10299 instances in total with captured action features for 30 participants. There are 6 actions in total which serve as labels for clustering. The identity of each person is used as the PSV value.

Following Bera et al. (2019) we choose three tabular datasets for complete comparison: 1) Census data with 5 attributes ("age", "fnlwgt", 'education-num", "capitalgain", "hours-per-week") and binary PSV gender, we set whether income exceeds 50K as the clustering label; 2) Bank data with 3 attributes ("age", "balance", "duration-of-account") and binary PSV marital, we set whether a client will subscribe a term deposit as the label; 3) Credit data with 14 features and binary PSV marital, we set whether the cardholder will make a payment as the label.

To measure the clustering quality for deep fair clustering and other baselines, we use both clustering accuracy (ACC) (Xu et al., 2003; Yang et al., 2010) and normalized mutual information (NMI) metrics. To evaluate the fairness, we use the balance measure defined in equation 2. For all those three measures, higher values indicate better performance. For the deep clustering baselines, we use DEC (Xie et al., 2016) as a representative method for centroid-based clustering and IMSAT (Hu et al., 2017) for discriminative clustering approach. For fair clustering algorithms, we choose the scalable fair clustering algorithm (Backurs et al., 2019) and the fair algorithms for clustering (Bera et al., 2019). For deep fair clustering baselines, we compare our work with the latest work (Li et al., 2020) and the geometric-based fair clustering (Wang & Davidson, 2019). For our own approach, we use two convolutional layers followed by batch normalization and pooling for visual data and fully connected layers for tabular data. For a fair comparison with non-deep baselines, we use pre-trained auto-encoder's features like (Li et al., 2020). For the LP solver we use the Gurobi optimizer. More details about datasets and experimental setup are given in Appendix.

### 5.2 EVALUATION

Fair clustering results on high dimensional data: as shown in the Table 1, traditional fair clustering algorithms achieve good fairness results especially ScFC which returns guaranteed fair clusters. However the clustering performance is not good as deep clustering methods due to the lack of representation learning. Both DEC and IMSAT achieve reasonable clustering results but poor balance, this shows the unfairness of existing deep clustering models which motivates our adding fairness rules. Comparing our results with the recent deep fair clustering works (Wang & Davidson, 2019; Li et al., 2020) we can see that our approach consistently outperforms these two in terms of both clustering performance and fairness. Note we report both the deep model's results and the ILP's output in the last iteration. We observe that our deep clustering model's predictions almost converge to the final assignments solved from our ILP module. Observing the ground truth results in Table 1

---

[1]In our experimental work we use the fair clustering data sets used by earlier work for comparison.

Table 1: Comparison of clustering and fairness performance on MNIST-USPS, Reverse-MNIST and HAR. HAR consists of *multi-state PSV* that baselines with dashes are not applicable. Bold results are the best results among all the baselines except the guaranteed fairness results which are marked with blue. Note we report our average performance results after 10 trials and the term *optimal* refers to the clustering giving the ground truth labels and the corresponding balance.

| | MNIST-USPS | | | Reverse-MNIST | | | HAR | | |
|---|---|---|---|---|---|---|---|---|---|
| Methods | ACC | NMI | Balance | ACC | NMI | Balance | ACC | NMI | Balance |
| Ground Truth (Optimal) | 1.000 | 1.000 | 0.120 | 1.000 | 1.000 | 1.000 | 1.000 | 1.000 | 0.458 |
| DEC Xie et al. (2016) | 0.586 | 0.686 | 0.000 | 0.401 | 0.480 | 0.000 | 0.571 | 0.662 | 0.000 |
| IMSAT Hu et al. (2017) | 0.804 | 0.787 | 0.000 | 0.525 | 0.630 | 0.000 | 0.812 | 0.803 | 0.000 |
| ScFC Backurs et al. (2019) | 0.176 | 0.053 | 0.120 | 0.268 | 0.105 | 1.000 | – | – | – |
| FAlg Bera et al. (2019) | 0.621 | 0.496 | 0.093 | 0.295 | 0.206 | 0.667 | 0.642 | 0.618 | 0.420 |
| Wang & Davidson (2019) | 0.725 | 0.716 | 0.039 | 0.425 | 0.506 | 0.430 | 0.607 | 0.661 | 0.166 |
| DFCV Li et al. (2020) | 0.825 | 0.789 | 0.067 | 0.577 | 0.679 | 0.763 | – | – | – |
| Fair-DEC (Deep Model) | 0.783 | 0.752 | 0.118 | 0.539 | 0.614 | 0.937 | 0.680 | 0.725 | 0.458 |
| Fair-DEC (ILP) | 0.774 | 0.746 | 0.120 | 0.509 | 0.590 | 1.000 | 0.668 | 0.705 | 0.653 |
| Ours Result (Deep Model) | **0.939** | **0.876** | 0.119 | **0.589** | **0.690** | 0.946 | **0.862** | **0.845** | 0.468 |
| Ours Result (ILP) | 0.936 | 0.867 | 0.120 | 0.583 | 0.680 | 1.000 | 0.842 | 0.827 | 0.653 |

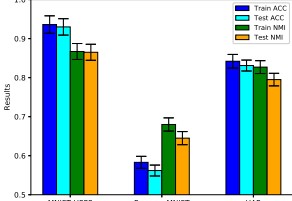

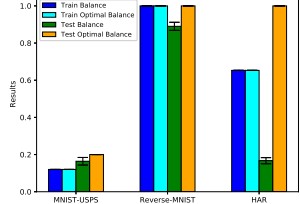

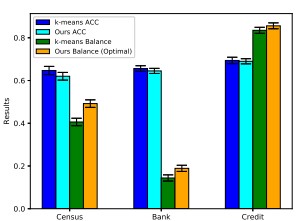

(a) Predictive Clustering Results     (b) Predictive Clustering Fairness     (c) Tabular Datasets Results

Figure 2: Experimental results on novel predictive clustering settings and classic tabular datasets .

we can see the fairness rules can be seen as *positive guidance* to improve the clustering performance, our approach is shown to be able to learn from this guidance and improve fairness as well as accuracy.

Predictive clustering results: here we evaluate our method's ability to make predictions on test data *without* PSV information which is a new setting in the fair clustering literature. That is we have already clustered a data set with PSV values and are now making predictions using the model learnt. This is particularly important for practitioners who are, for instance, deploying models on the web (where individuals are reluctant to share PSV information) and we see our results in Figure 2. Our approach performs consistently in both train and test sets. One exception is that the test balance in HAR is much lower than the training balance, we hypothesize this is due to different distributions between training and test set, the optimal test balance is 1 while the optimal training balance is 0.653.

Classic tabular datasets: we evaluate our approach on tabular data sets and present the results in Figure 2 (c). We find that our model achieves similar clustering results as k-Means on tabular data with human-defined semantic features. Meanwhile, we notice that k-Means algorithm achieves good fairness results when the number of clusters is correctly set as the ground truth number of classes. Finally, our model can achieve optimal balance with a slight loss in terms of clustering accuracy.

### 5.3 FURTHER ANALYSIS ON OUR MODEL

Here we explore our model's flexibility in satisfying fairness requirements and better understand its performance by feature space visualization, parameter sensitivity and empirical convergence study.

Feature space visualizing: to understand how our model learns a fair representation, we have applied t-SNE (Van der Maaten & Hinton, 2008) to visualize the feature space of MNIST-USPS during different training epochs in Figure 3. The initial model is trained with clustering objectives which yield unfair results, once we introduce fairness signals the red instances start to move to different clusters. Meanwhile, we observe our learned representations maintain good clustering properties.

Tuning the weight of fairness objective: we experiment on the choices of hyper-parameter $\beta$ which controls the weight of the fairness objective and report the clustering results in Figure 4. It is straightforward to see from (a) and (c) that as $\beta$ increases, the training balance increases. Meanwhile,

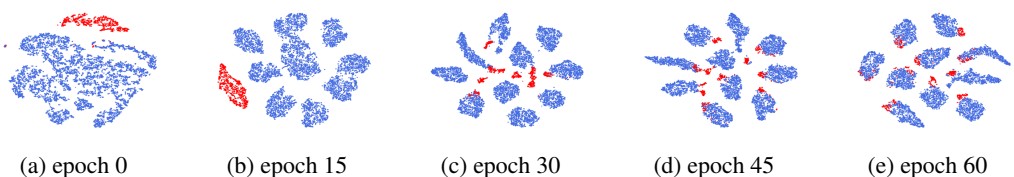

Figure 3: t-SNE visualization of learned embedding, color red and blue indicate different PSV values.

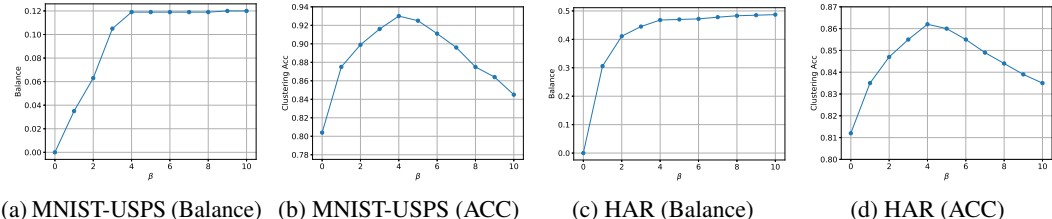

Figure 4: Sensitivity analysis of hyper-parameter $\beta$ which serves as the weights for fairness objective.

based on (b) and (d) we can find the ACC goes up and down as $\beta$ increases. Our previous result shows that the fairness constraints can serve as positive guidance for both MNIST-USPS and HAR. That is why the clustering accuracy goes up when we increase $\beta$ from 0. But we also observe that with a very large $\beta$ the clustering accuracy will drop. We hypothesize this is because the fairness objective dominates the overall objective so that the impact of clustering objective is hindered. As balance can be tracked during the training process for free, our insight for selecting hyper-parameter $\beta$ is to pick the smallest $\beta$ that achieves satisfying balance results.

Results on flexible fairness constraints with MNIST-USPS: here we explore how relaxing the optimal fair condition defined as $\rho_t$ in equation 9 produces flexible constraints. We now require the fairness requirement to be in the interval $[\rho_t * (1 - \epsilon), \rho_t * (1 + \epsilon)]$. In Figure 5 (a), we can see a larger relaxation degree $\epsilon$ leads to a lower balance which as expected; since the fairness signals can serve as positive guidance for clustering in MNIST-USPS, we observe the ACC and NMI are decreasing with larger $\epsilon$. Allowing flexible constraints are important as the fairness rules vary across regions.

Empirical convergence analysis: to investigate the smoothness of learning with clustering and fairness objectives together, we present the learning curves of overall training loss and the balance results in Figure 5 (b) and (c). We can see from the plots that our model's overall training loss drops quickly and converges after 50 epochs. Meanwhile, our model's balance result also converges after 50 epochs.

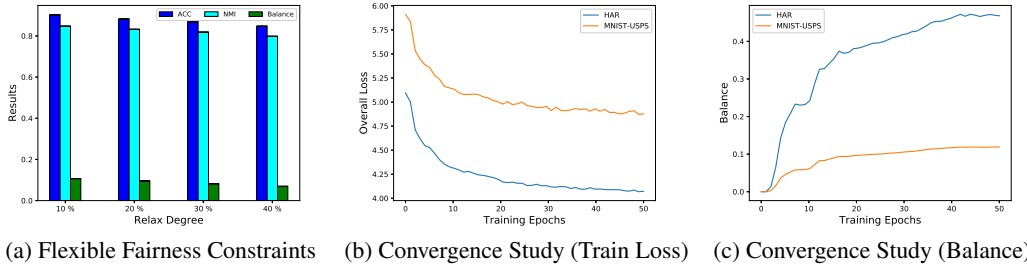

Figure 5: Flexible fairness constraints experiments on MNIST-USPS in (a); visualizing the learning curves of training loss and fairness measured by the balance on HAR and MNIST-USPS in (b, c).

## 6    CONCLUSION

In this paper, we explore the novel direction of adding fairness into deep clustering. This is a challenging problem given the end-to-end deep learning setting which does not facilitate pre-processing into fairlets and the need for scalability to large data sets. We formulate a group level measure of fairness as an integer linear programming problem and show the problem can be solved efficiently due to total unimodularity (Theorem 4.2). We then add this solver into a deep learner and show that our formulation works with multi-state sensitive attributes as well as flexible fairness constraints that can occur in real-life applications. Extensive experiments demonstrate the strong performance of our approach and an in-depth analysis including feature space visualization, hyper-parameter tuning, model convergence analysis, and investigating flexible fair constraints shows its versatility.

ETHICS STATEMENT

Our work is proposed to ensure the fairness of deep clustering algorithms. One advantage of our work is that users can define their own group-level fairness criterion, as shown in the flexible fairness rules experiments. Another advantage is that the learned model can be used for out-of-sample predictions of new instances which may not have sensitive attributes (PSVs) available. However, our flexible design will also allow malicious users to pass the "unfair constraints" into our ILP fairness objective, which leads to artificially unfair results. Therefore, we recommend that the users output both the fairness requirements within the ILP objective and the deep clustering model's results to ensure the correct use of our algorithm.

REPRODUCIBILITY STATEMENT

After the reviewing process, our work will be open-sourced, allowing researchers to conduct deep clustering tasks on both tabular and image datasets with group-level fairness rules. Moreover, we have included our source code in the supplementary package and also added two extra sections about baselines implementations and our model's selected hyper-parameters in the Appendix.

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

# A  THEOREMS AND PROOFS

**Lemma A.1.** *(Lemma 3.1). The optimal balance can be achieved only when all the clusters have the same balance. Formally, $\forall i, j \in \{1, 2, ..., K\}$: $balance(C_i) = balance(C_j)$.*

*Proof.* The proof is by contradiction, we assume the optimal balance can be achieved when not all clusters have the same balance. Let cluster $C_i$ have the largest balance as $b_{max}$ and cluster $C_j$ has the smallest balance as $b_{min}$. Denote the number of $T$ different groups' instances in $C_i$ as $\{C_{i1}, C_{i2}, \ldots, C_{iT}\}$, similarly the composition for $C_j$ as $\{C_{j1}, C_{j2}, \ldots, C_{jT}\}$. Assume the balance for $C_i$ is achieved by $b_{max} = \frac{C_{i\alpha}}{C_{i\beta}}$, the balance for $C_j$ is achieved by $b_{min} = \frac{C_{j\gamma}}{C_{j\theta}}$. Now we discuss two possible cases:

**Case 1:** If $\alpha = \gamma$ and $\beta = \theta$. Based on our definition we have $\frac{C_{i\alpha}}{C_{i\beta}} > \frac{C_{j\alpha}}{C_{j\beta}}$. Let $r = \frac{1}{2}(\frac{C_{i\alpha}}{C_{i\beta}} + \frac{C_{j\alpha}}{C_{j\beta}})$, we can move $\epsilon$ instances which belong to group $\alpha$ from $C_i$ to $C_j$ to achieve the higher balance $r$. This can be done by setting $\frac{C_{i\alpha} - \epsilon}{C_{i\beta}} = \frac{C_{j\alpha} + \epsilon}{C_{j\beta}} = r$.

**Case 2:** If $\alpha \neq \gamma$ or $\beta \neq \theta$. Based on the balance definition we have $\frac{C_{i\gamma}}{C_{i\theta}} \geq \frac{C_{i\alpha}}{C_{i\beta}}$. Based on our definition we have $\frac{C_{i\alpha}}{C_{i\beta}} > \frac{C_{j\gamma}}{C_{j\theta}}$. Thus we have $\frac{C_{i\gamma}}{C_{i\theta}} \geq \frac{C_{i\alpha}}{C_{i\beta}} > \frac{C_{j\gamma}}{C_{j\theta}}$. Similar as in case 1 we can move $\epsilon'$ instances which belong to group $\gamma$ from $C_i$ to $C_j$ to achieve higher balance $r' = \frac{1}{2}(\frac{C_{i\gamma}}{C_{i\theta}} + \frac{C_{j\gamma}}{C_{j\theta}})$.

In both cases, we can swap some instances between clusters to increase the final balance. This contradicts our assumption and completes our proof.  □

**Theorem A.2.** *(Theorem 3.2). To achieve optimal balance value for multi-state protected variables, we must satisfy the condition: $P(x \in G_t | x \in C_k) = \rho_t$ which is precisely the optimal fairness value for equation[2] 2.*

*Proof.* Given the condition $P(x \in G_t | x \in C_k) = \rho_t$, let $G_{min}$ be the smallest protected group and $G_{max}$ be the largest protected group, the largest balance we can achieve is $|G_{min}|/|G_{max}|$. The proof by contradiction assumes these exists a solution where all the clusters have the same balance (based on lemma A.1) and have balance $\alpha > |G_{min}|/|G_{max}|$. For each cluster we sort each protected group based on their size in an increasing order. We use $C_{i1}^*, C_{i2}^* \ldots C_{iT}^*$ to denote the size of the sorted groups in cluster $i$. Obviously we have $\alpha = \frac{C_{i1}^*}{C_{iT}^*}$ for any cluster $i$. Now we sum up the smallest group among all the $K$ clusters as $S$: $S = \sum_{i=1}^{K} C_{i1}^*$. Similarly we can calculate the sum of the largest group among all the clusters as: $\sum_{i=1}^{K} C_{iT}^* = \frac{1}{\alpha} \sum_{i=1}^{K} C_{i1}^* = \frac{S}{\alpha}$. Consider the allocation of the smallest group $G_{min}$ we have the following inequality: $|G_{min}| \geq \sum_{i=1}^{K} C_{i1}^* = S$. Similarly we have $|G_{max}| \leq \sum_{i=1}^{K} C_{iT}^* = \frac{S}{\alpha}$. By combining previous two inequalities we have $\alpha |G_{max}| \leq S \leq |G_{min}|$ which means $|G_{min}|/|G_{max}| \geq \alpha$. This contradicts with our initial assumption that there exists a $\alpha$ which is larger than $|G_{min}|/|G_{max}|$. Thus we complete the proof.  □

**Proof Total Unimodularity of Constraint Matrix** We know that if a constraint matrix of an ILP is totally unimodular (TU) then we can solve the problem using an LP (linear program) solver and the solution will still be integral Schrijver (1998). Using an LP solver will largely reduce the running time and Vaidya (1989) has shown that the running time for LP is polynomial in the input size.

In the above proposed constraints, there are $NK$ unique regular variables ($N$ instances and $K$ categories). To construct the constraint matrix $C$ which encodes constraint 7, constraint 8 and constraint 9, we will use $NK$ regular variables. Matrix $C$ has $T+1$ rows (the first $T$ rows correspond to the fairness constraints in equation 7 and last row corresponds to constraints in equation 9) and $N + K$ columns. Note the first $K$ columns of the last row are set to $0$ and the last $n$ columns of first $T$ rows are set to $0$. In matrix $C$, each entry of $C$ is from $\{-1, 0, 1\}$. Moreover, each column only has one non-zero element. This is because: (1) for constraints set in equation 7, each instance only belongs to one protected group, (2) for constraints set in equation 9, there is only one row vector with $K$ elements as $1$ to ensure the valid assignment.

---

[2] All the equations referenced in appendix are in the main paper.

**Lemma A.3.** *(Lemma 4.1).* ***TU Identity*** *(Schrijver, 1998). Let $C$ be a matrix such that all its entries are from $\{0, 1, -1\}$. Then $C$ is totally unimodular, i.e., each square submatrix of $C$ has determinant $0$, $1$, or $-1$ if every subset of rows of $C$ can be split into two parts $A$ and $B$ so that the sum of the rows in $A$ minus the sum of the rows in $B$ produces a vector all of whose entries are from $\{0, 1, -1\}$.*

**Theorem A.4.** *(Theorem 4.2). The matrix $C$ formed by the coefficients of the constraints used to encode our proposed constraints from equation 7, 8 and equation 9 is totally unimodular.*

*Proof.* We consider any subset $F$ of rows in $C$. We will show that $F$ can be partitioned into two sets $A$ and $B$ to satisfy the condition in lemma A.3. Our partitioning scheme is as follows: the first row of $F$ is put into $A$ and the remaining rows are put into $B$. Let row vectors $SA$ and $SB$ denote the sums of the rows within $A$ and $B$ respectively. It is clear that elements in $SA$ and $SB$ are from $\{1, 0, -1\}$ because each column only has one non-zero element. Now we show all the elements in $SA - SB$ are from $\{1, 0, -1\}$. Firstly there will be only one non-zero element in $SA$, and we denote the column which has a non-zero element as $r$. If column $r$ belongs to the fairness constraints (from column 1 to column $N$ in $C$), then the column $r$ within $SB$ will be 0 since one instance only belongs to one protected group. If the column $r$ belongs to the valid assignment constraints (from column $N + 1$ to $N + K$ in $C$) then column $r$ within $SB$ will be 0 since those elements are filled with 0. Thus column $r$ of $SA - SB$ is still from $\{1, 0, -1\}$. The non-zero elements in other columns will change their sign but still from $\{1, 0, -1\}$. This completes the proof. $\square$

## B  BASELINES IMPLEMENTATIONS

For the k-means baseline, we choose to use the k-means++ implementation from scikit-learn library. For scalable fair clustering work (Backurs et al., 2019), we use the code from their author's github repository (`https://github.com/talwagner/fair_clustering`). For deep embedding clustering (Xie et al., 2016) we use the implementations from the public github repositories (`https://github.com/vlukiyanov/pt-dec`) and for IMSAT (Hu et al., 2017) we use the code from their author's github repository (`https://github.com/weihua916/imsat`). For Wang and Davidson's deep fair clustering work (Wang & Davidson, 2019), we have contacted the author to get their code re-run the experiments under our setting. For the DFCV (Li et al., 2020) approach, we directly excerpt the experimental results from their original paper.

## C  HYPER-PARAMETERS AND ARCHITECTURES OF OUR APPROACH

For the digit datasets like reverse-MNIST and MNIST-USPS, we use a 2-block convolutional architecture with first block consisting of a convolutional layer with 8 ($5 \times 5$) filters and second block consisting of 4 ($5 \times 5$) filters. Each block is followed with BatchNorm, ReLU activation and max pooling layers. The second block is finally connected to a fully connected layer with 10 units. For the HAR data we use a stacked fully connected neural network with intermediate layers as $d - 1200 - 1200 - k$, note $d = 561$ is the total number of features in HAR and $k = 6$ is the number of clusters. Moreover, we have listed the hyper-parameters that we used for these three data sets in Table 2. In unsupervised learning, it is not straightforward to determine hyper-parameters by cross-validation. Hence, we fixed the hyper-parameters (weight decay $\alpha$ and augmentation term $\gamma$) across all the datasets; we tuned the hyper-parameter $\beta$ based on the training set's balance value as mentioned in the main paper to achieve satisfying fair results.

Table 2: Deep-clustering data: hyperparameter summary

|  | HAR | reverse-MNIST | MNIST-USPS |
|---|---|---|---|
| Learning rate | 0.0002 | 0.0005 | 0.0005 |
| Batch size | 256 | 256 | 256 |
| Train epochs | 60 | 80 | 50 |
| Weight decay $\alpha$ | 0.0001 | 0.0001 | 0.0001 |
| Parameter $\beta$ | 4.00 | 6.00 | 4.00 |
| Parameter $\gamma$ | 1.00 | 1.00 | 1.00 |

Table 3: Traditional tabular data: hyperparameter summary

|  | Census | Credit | Bank |
| --- | --- | --- | --- |
| Learning rate | 0.0002 | 0.0002 | 0.0005 |
| Batch size | 256 | 256 | 256 |
| Training epochs | 30 | 50 | 30 |
| Weight decay $\alpha$ | 0.0001 | 0.0001 | 0.0001 |
| Parameter $\beta$ | 4.00 | 5.00 | 4.00 |
| Parameter $\gamma$ | 1.00 | 1.00 | 1.00 |

For the traditional fair clustering data sets, we use a stacked fully connected neural network with intermediate layers as $d - 50 - 50 - k$, note the total number of features $d$ for Census, Credit and Bank are $5, 3, 14$ and $k = 2$ is the number of clusters. Moreover, we have listed the hyperparameters that we used for these three data sets in Table 3. We set the same values for hyper-parameters $\alpha$ and $\gamma$. Since these tabular data sets only have limited features we observe that fewer training epochs are needed to train our model.

## D  EXTRA EXPERIMENTAL RESULTS

In this section we first report the run time results in Table 4. Our codes will be available on GitHub for public use after the reviewing process.

### RUNTIME ANALYSIS

We implement our algorithm in Python 3.6 and run our experiments on a PC with a 3.3 GHz Intel Core i9 Processor and 1 GeForce RTX 2080 TI graphic card. We use Gurobi [3] for solving all the ILP problems. We report the runtime results in Table 4. We can observe that solving the ILP via LP is not very time-consuming, the introduction of our fairness objective does add more computations compared with the non-fair version but the overall time cost is still acceptable.

Table 4: We report the average running time (second) over 10 trials.

|  | ILP solver | Train with fairness objective | Train w/o fairness objective |
| --- | --- | --- | --- |
| HAR | 87.20 | 130.81 | 12.49 |
| reverse-MNIST | 389.71 | 813.26 | 218.90 |
| MNIST-USPS | 201.56 | 434.18 | 127.76 |
| Census | 32.65 | 96.13 | 22.54 |
| Credit | 53.82 | 151.13 | 34.28 |
| Bank | 5.12 | 12.49 | 3.57 |

### CLUSTERING RESULTS VISUALIZATION

Figure 6 and 7 shows randomly selected instances of the MNIST-USPS that clustered via our proposed fair clustering approach and the deep clustering approach (IMSAT). We observe that in the IMSAT's results most of the instances from USPS are grouped to one cluster which lead to 0 balance. Instead, our approach successfully separated them into different clusters to achieve fairness goals.

---

[3]https://www.gurobi.com/

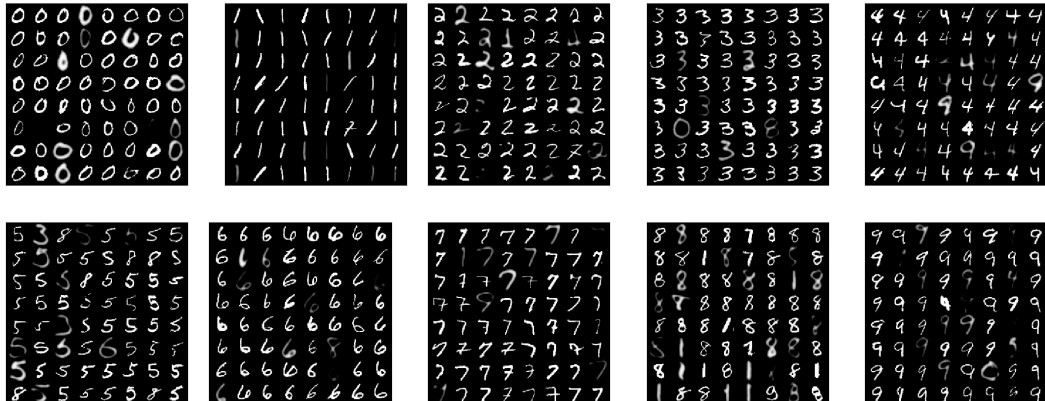

Figure 6: Sampled instances of MNIST-USPS discovered using our fair clustering approach.

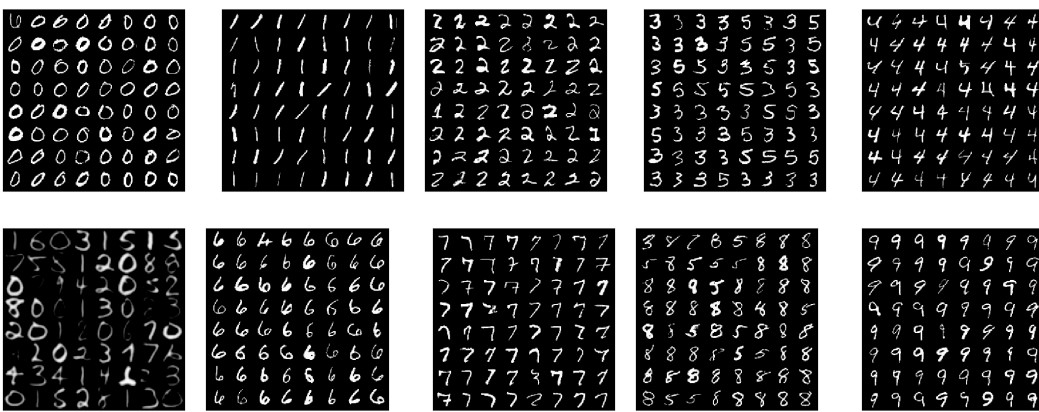

Figure 7: Sampled instances of MNIST-USPS discovered using IMSAT.

