# OpenReview forum: "Deep Fair Discriminative Clustering"
_ICLR.cc/2022/Conference — ICLR 2022 Submitted_

### Official Review · Reviewer_tDB1 · 2021-10-30

**Correctness:** 3
**Technical Novelty And Significance:** 2
**Empirical Novelty And Significance:** 3
**Recommendation:** 5
**Confidence:** 4

**Main Review:**

## Pros
1) The proposed fair clustering algorithm shows the equivalence between the fairness measure defined in eqn 3 and the balance measure.
2) The experimental results seem satisfactory, especially the balance measure.
3) This paper is well-organized and easy to follow.

## Cons
1)  As Section 4.1 is simply the introduction of the backbone network, maybe it should be an individual section. In my opinion, the major contribution is to use the fair approximation as pseudo-labels. It will be more convincing to conduct experiments on various backbone networks.
2) In P5, the authors claim that "Let the rounded version of current assignment $Y$ as $Y'$ then the size of cluster $C_j$ ... ". But what is the *rounded version* (formally)? Does it mean the hard version of $Y$ as $Y$ is a continuous variable?
3) In Table 1, the balance scores of DEC and IMSAT are 0. It means that some samples from USUS are not assigned to a certain cluster (on MNIST-USPS). It is abnormal since ACC and NMI are still high. Although these two models are non-fair clustering methods, it might be inappropriate to mark the balance score as 0 simply.
4) There are some typos and inappropriate formulation:
- P5: "that *satisfy* our optimal fairness condition"
- P5, eqn 8: $*$ -> $\times$ or $\cdot$. What's more, the authors should emphasize that $y_i$ is a row vector.
- P5, eqn 11: the superscript is missing.


**Summary Of The Paper:**

This paper proposes a fair clustering algorithm that uses DL models to map the data into deep representations. The authors also show the equivalence between the practical fairness measure and the balance measure. The algorithms can be concluded as two steps:
1) find fair assignments $\hat y$ based on $y$;
2) tune the latent representations and $y$ according to the pseudo-label $\hat y$.

**Summary Of The Review:**

Please see above.

---

> ### Author Response · Authors · 2021-11-22
> **Response to Reviewer tDB1**
>
> We thank the reviewer's helpful suggestions for our work and updated our paper based on comments. We understand the reviewer's concerns and hope the following answers provide a satisfying response.
>
> **As Section 4.1 is simply the introduction of the backbone network, maybe it should be an individual section. In my opinion, the major contribution is to use the fair approximation as pseudo-labels. It will be more convincing to conduct experiments on various backbone networks.**
>
> Thanks for your suggestion, we changed the title of this subsection into a review of the base clustering algorithm. As for your suggestion on conducting experiments on various backbone networks, we did a new set of experiments by replacing the current based clustering with DEC [1]. Note DEC is a popular centroid-based deep clustering algorithm which also outputs a probabilistic clustering assignment. We update Table 1 in our experimental section with the new results of fair-DEC. The empirical results have shown that fair-DEC also achieved good  fairness results which outperforms the existing baselines.This shows our methods of adding fairness has wider scale applicability than claimed in the original paper.
>
> **In P5, the authors claim that "Let the rounded version of current assignment  $Y$ as $Y^{'}$ then the size of cluster $C_j$ ... ". But what is the rounded version (formally)? Does it mean the hard version of  $Y$ as  $Y$ is a continuous variable?**
>
> Thanks for pointing it out, the reviewer's guess is correct. We round the probabilistic assignment $Y$ as binary uni-modal assignments $Y^{'}$ by assigning the cluster with largest probability (e.g., [0.2, 0.5, 0.3] -> [0, 1, 0]). We have updated our paper to make it clear.
>
> **In Table 1, the balance scores of DEC and IMSAT are 0. It means that some samples from USPS are not assigned to a certain cluster (on MNIST-USPS). It is abnormal since ACC and NMI are still high. Although these two models are non-fair clustering methods, it might be inappropriate to mark the balance score as 0 simply.**
>
> Thanks for noting this, we have added one section in the appendix to visualize part of the clustering results for MNIST-USPS. Figure 6 and 7 shows randomly selected instances of the MNIST-USPS that clustered via our proposed fair clustering approach and IMSAT. We observe that for IMSAT all the instances from USPS are grouped to one cluster and our approach successfully separated them into different clusters to achieve fairness goals. The reason that ACC and NMI are still high for IMSAT is because the size of the USPS dataset is small so that the clustering performance won't decrease too much.
>
> [1] Xie, Junyuan, Ross Girshick, and Ali Farhadi. "Unsupervised deep embedding for clustering analysis." International conference on machine learning. PMLR, 2016.

---

### Official Review · Reviewer_py3W · 2021-11-01

**Correctness:** 3
**Technical Novelty And Significance:** 2
**Empirical Novelty And Significance:** 2
**Recommendation:** 3
**Confidence:** 3

**Main Review:**

The own contributions of this paper are unclear. The introduced fairness notion is simply a generalization of the previous proposed measure of fairness allowing for protected status variable (PSV) with multi-state instead of binary-state. The integer linear programming formulation is then plug into a probabilistic discriminative clustering model proposed in 2017 for deep fair discriminative clustering. The own contributions therefore remain unknown. It is a stretch to argue that the authors should not use "we" as the subjects when describing the base clustering model and others' methods which mystify their own contributions.

Relevantly, multi-state PSV is stated as their learning target while 2/3 datasets are binary PSV. In addition, the improvements over baselines are marginal or no improvement and flexible fairness constraint is not completely supported by the experimental results notably the ACC part. The authors should consider tone down relevant claims.

This study is not well motivated either. The drawbacks of existing studies and how these limitations are addressed by this work should have discussed to justify the necessary of this work. The rationals of selecting the base clustering model and evaluation metrics also lack of more detailed discussion.

**Summary Of The Paper:**

The authors propose to combine deep learning and discrete optimization to learn fair representation for clustering. To this end, a generalized notion based on the previous defined balance measure is introduced, based on which an integer linear programming problem is defined and is further added into an existing deep learner for fair deep clustering.

**Summary Of The Review:**

Lack of own contribution, proper motivation and sufficient justification of the proposed method.

---

> ### Author Response · Authors · 2021-11-22
> **Response to Reviewer py3W**
>
> Thank you for your feedback. It seems there are some misunderstandings of our paper's contributions and the corresponding experimental analysis that are easy to address . We hope the reviewer will re-evaluate our work. We address your questions below in detail.
>
> **“The own contributions of this paper are unclear. The introduced fairness notion is simply a generalization of the previous proposed measure of fairness allowing for a protected status variable (PSV) with multi-state instead of binary-state. The integer linear programming formulation is then plug into a probabilistic discriminative clustering model proposed in 2017 for deep fair discriminative clustering”**
>
> At the end of the introduction (middle of page 2), we list our contributions. We reiterate them below in the context of the reviewers’ comments.
>
> In our first claimed contribution, we introduced a general notion of fairness and proved that optimizing the general notion of fairness is equivalent to optimizing the balance measure for disparate impact previously used. Further, how to encode fairness in deep clustering is an understudied problem; the idea of generating pseudo fairness signals from ILP and then tuning the deep clustering representation via these pseudo labels is novel. It provides the opportunity for encoding flexible fairness constraints and for multi-state PSVs. Moreover, to make the ILP solver feasible for deep clustering tasks, we also show how to relax the constraint matrix to be totally unimodular so that the sub-problem can be efficiently solved by an LP solver (Theorem 4.2). Another contribution which the reviewer missed is that our model achieved state-of-the-art fairness results by surpassing the existing deep fair clustering works in a large margin. Based on Reviewer tDB1's suggestions, we experimented with different clustering backbones and reported the results in Table 1. Our approach is demonstrated to improve the fairness results under different clustering backbones significantly.
>
> **Relevantly, multi-state PSV is stated as their learning target while 2/3 datasets are binary PSV. In addition, the improvements over baselines are marginal or no improvement and flexible fairness constraint is not completely supported by the experimental results notably the ACC part.**
>
> We use classic data sets that other works use most of which focus on the binary PSV setting; this is unavoidable. We do not understand the claim that our improvements are “marginal or no improvement”. Based on Table 1, we have significant improvement.  For the multi-state PSV problem (see column HAR in Table 1), our proposed model outperforms existing deep fair clustering in terms of fairness.  As for the binary PSV problems, our proposed approach also significantly improves the balance (fairness) which beats all the deep fair clustering baselines. For the flexible constraints part, the fairness constraints in MNIST-USPS can serve as positive guidance which can help increase the clustering performance.  The more we relax the constraints, the lower balance and clustering performance we will get.
>
> **The drawbacks of existing studies and how these limitations are addressed by this work should have discussed to justify the necessary of this work.**
>
> The drawbacks of previous works are stated in the related work section and the fourth paragraph of introduction. In our experimental section, we show our model significantly improves the fairness performance against existing deep fair clustering works and largely outperforms the traditional fair clustering works in terms of clustering quality. Besides that we show that our proposed approach can work in some novel clustering settings including predictive clustering, supporting flexible fairness rules.
>
> **The rationals of selecting the base clustering model and evaluation metrics also lack of more detailed discussion.**
>
> We stated the reason why we prefer [1] as our base clustering model at the end of section 4.1. As for the evaluation metrics, for deep clustering performance we adopt the commonly used clustering ACC and NMI metrics and cite the sources of these two measurements in section 5.1. For the fairness measurement we use the balance measure which follows all the previous fair clustering works.
>
> [1] Hu, Weihua, et al. "Learning discrete representations via information maximizing self-augmented training." International conference on machine learning. PMLR, 2017.

---

### Official Review · Reviewer_Um5H · 2021-11-01

**Correctness:** 3
**Technical Novelty And Significance:** 2
**Empirical Novelty And Significance:** 2
**Recommendation:** 5
**Confidence:** 4

**Main Review:**

Strengths:
1. This paper explores the fairness problem in deep clustering, and presents a general notion of fairness for both binary and multi-state protected status variables, which is proved to be equivalent to optimize the general balance measure for disparate impact.

Weaknesses:
1. The paper said that it can be fairer in clustering without the sensitive attributes. However, what is sensitive attributes in a data set? The definition is not clear. How do we judge whether attribute is sensitive? The authors think the clustering result is not fair, if the sensitive attributes are used in clustering process. Why do we delete these sensitive attributes and cluster the data set without sensitive attributes? Why doe we need the fairness constraints for clustering task? The authors should provide the clear explanation.
2. In the proposed algorithm, the authors first get pseudo labels y on a data set with sensitive attributes by a deep clustering model. Furthermore, they use fairness constraints to refine the pseudo labels to get \hat{y}. Finally, they use \hat{y} as supervised information to train a deep model. That make me confuse: why do the authors directly add the fair constraints to the objective function or problem of the deep model? Why are the fairness constraints only used as a preprocess of pseudo labels?
3. In the experiments, these tested datasets do not include sensitive attributes. The authors should provide an application scenario to prove the performance of the proposed algorithm.


**Summary Of The Paper:**

This paper proposes a novel deep fair clustering framework which addresses the problem about producing guaranteed fair predictions on clustered data with PSVs and making out-of-sample fair predictions for data without PSV.

**Summary Of The Review:**

The present paper presents a framework to ensure the fairness of deep clustering algorithms. The paper is well organized and the motivation of this paper is clear. However, some definitions and roles about sensitive attributes are not clear. Besides, these tested datasets do not include sensitive attributes. The experiments are not enough.

---

> ### Author Response · Authors · 2021-11-22
> **Response to Reviewer Um5H**
>
> Thanks for your feedback. We hope the following point-by-point response addresses your concerns.
>
> **However, what is sensitive attribute in a data set? The definition is not clear. How do we judge whether attribute is sensitive?**
>
> Note [1] uses the notion of the sensitive attribute which is equivalent to the protected status variable (PSV) in our work. Sensitive attributes are features that may not be used as the basis for decisions. Sensitive attributes could be chosen because of legal mandates or because of organizational values. Some common sensitive attributes include race, religion and gender.
>
> **The authors think the clustering result is not fair, if the sensitive attributes are used in the clustering process. Why do we delete these sensitive attributes and cluster the data set without sensitive attributes? Why doe we need the fairness constraints for clustering task?**
>
> In real-world data, there are many normal features which are associated/surrogates with/for the sensitive attributes (e.g., the height/weight can be associated with gender). Thus simply removing the sensitive attributes and running the clustering algorithm cannot guarantee fair partitions. This is the difference between disparate treatment and disparate impact. Therefore, we need fairness constraints to help clustering models learn a fair representation (de-biasing the surrogate features for sensitive attributes) to decorrelate the clustering decisions with sensitive attributes.
>
> **Why do the authors directly add the fair constraints to the objective function or problem of the deep model? Why are the fairness constraints only used as a preprocess of pseudo labels?**
>
> In the fourth paragraph of our introduction section, we showed that previous works [1, 2] that directly added the fairness objectives to the deep model performed not very well in terms of fairness. As for our proposed model, we first encode fairness constraints into the ILP problem to generate fair pseudo-labels which are guaranteed to be fair and then tune the latent representation according to the guaranteed fair signals. Another benefit of encoding the constraints in this way is that we can allow for flexible fairness constraints. Further, we have empirically shown in our experimental analysis (Table 1) that our proposed approach is much fairer (measured by balance) than [1, 2].
>
> **In the experiments, these tested datasets do not include sensitive attributes. The authors should provide an application scenario to prove the performance of the proposed algorithm.**
>
> In section 5.1, we introduced the characteristics of all the fair clustering data sets used by earlier work. Even though the sensitive attributes are not included in these datasets, the normal features can still be correlated to those sensitive attributes. Due to these associated normal features, we see that deep clustering algorithms like DEC and IMSAT achieve poor fairness results in Table 1.
>
> [1] Li, Peizhao, Han Zhao, and Hongfu Liu. "Deep fair clustering for visual learning." Proceedings of the IEEE/CVF Conference on Computer Vision and Pattern Recognition. 2020.
>
> [2] Wang, Bokun, and Ian Davidson. "Towards fair deep clustering with multi-state protected variables." arXiv preprint arXiv:1901.10053 (2019).

---

### Official Review · Reviewer_5Bwe · 2021-11-05

**Correctness:** 3
**Technical Novelty And Significance:** 3
**Empirical Novelty And Significance:** 2
**Recommendation:** 6
**Confidence:** 4

**Details Of Ethics Concerns:**

No ethical concerns.

**Main Review:**

The paper is generally clear and well written, and the empirical results are impressive.

The paper could do a better job distinguishing itself from existing work in deep fair clustering, which also supports multiple-state PSVs (this point seems muddied at times, e.g., the second to last sentence of section two and bullet 2 in section 5 both seem to imply otherwise).  It seems the main advantage of the proposed method over existing methods (beyond better observed performance in experiments, which is motivating in itself) is that the method guarantees a fair clustering.  Unfortunately, it is not very clear how this method is able to guarantee a fair clustering, since fairness is a weighted term in the loss function, and the authors don’t seem to prove (or argue) that the method will converge to a fair clustering (I can certainly believe that it would, but perhaps it would be useful to help the reader see that this is the case), and the authors also do not address what happens when a fair clustering is not possible.  I suppose the fairness guarantee involves using the output from the ILP in the last iteration as the output, but this doesn’t seem clearly specified.

One limitation of this work seems to be that each person can only belong to one protected class, which doesn’t seem to be the case for (Bera et al., 19).

The importance of Theorem 3.2 (one of the stated contributions of the paper) is not very clear and could be emphasized.

I found the description of the constraint matrix a little confusing and would imagine a little more clarification could be helpful.  For example, why are 2NK variables used (rather than NK), and doesn’t the number of variables typically equal the number of columns in the constraint matrix?

The empirical results are impressive, though counter intuitive at times and could benefit from a little additional explanation (perhaps as the result of a bit more exploration).  For example, it would seem intuitive that taking a loss function and adding a term for fairness would cause the result to be worse w.r.t. the original loss function.  While neither accuracy nor NMI are the original loss function, they (especially NMI) seem more closely related than to the fairness term.  The authors note this and claim that fairness provides “positive guidance” for clustering performance, but this does not seem to be an explanation as much as a hypothesis (or, less generously, a re-statement of the observation).  A second example that could probably benefit from slightly more explanation is the observed results when setting the beta hyper parameter.  Why does beta=4 seem to be the sweet spot for all datasets and metrics?  Any intuition for this (if any could be gained) would be interesting to report.  A third example, “we observe the ACC and NMI are decreasing with larger ε”, seems very counter intuitive  (similar to the first example, but perhaps more so) and some explanation beyond “positive guidance” would be encouraging.

Minor comments:

“Fairness takes two primary forms: i) group-level fairness and ii) individual-level fairness.” Please consider citing, I have seen other characterizations of fairness.

“Different geographic regions place this tolerance at different levels.” Consider adding a citation.

“most of these algorithms evaluate their performance on low-dimensional tabular data and mainly study the problems with binary PSV.”  Most of and mainly seem like unfortunate hedges here.  Perhaps its worth pointing out which work evaluate their performance on low-dimensional tabular data and study the problems with binary PSV, and which do not.

“Finally, the base clustering model optimizes the clustering loss lC and lAug simultaneously” Doesn’t (Hu et al., 2017) optimize lC subject to constraints on lAug? (And isn’t this slightly different?)

“Let the rounded version of current assignment Y as Y’ ” I’m not sure I understand what is meant here (perhaps due to a typo resulting in a grammatical error).

The choice of datasets makes sense (in that they were used in prior work).  Nevertheless, using MNIST for fairness seems somewhat contrived and the results would seem more motivating on more reasonable data.

What is ““fnlwgt”?  It might be worth spelling out what the attributes mean/their possible values.

It seems like an overstatement to claim that by evaluating both accuracy and NMI the study is comprehensive (which means  including all or nearly all elements or aspects of something).

“Note we report both the deep model’s results and the final ILP’s results.”  What is meant here does not seem to be specified anywhere and was confusing to me (and I think resulted in my confusion elsewhere).  I think a sentence or two explanation could go a long way.

“ground-truth and the guaranteed fairness results which are marked with blue”. I guess I know what is meant here, but there is a whole row labeled ground truth and only some of the cells in that row are blue.

“This is particularly important for practitioners who are, for instance, deploying models on the web (where individuals are reluctant to share PSV information)”. If the train data came from a source where PSV information is available and the test data came from a source where it is not (e.g., the web, as described), is it reasonable to assume other changes in the population/resultant distribution are present as well?

“We hypothesize this is because the fairness objective dominates the overall objective so that the impact of clustering objective is hindered” What is meant here?  That training solely on the fairness objective does not lead to good clustering results?  Something else?


**Summary Of The Paper:**

This paper proposes a method for fair clustering (where a clustering is considered fair when each protected group is present in every cluster in the same proportion as in the population) using deep neural networks.  The method works by training a neural network for clustering using the deep clustering with virtual adversarial training approach proposed by (Hu et al., 2017) and then refining (retraining) the network by adding a fairness term to the loss function (where the fairness term is the cross entropy loss between the current networks prediction and the fair assignments obtained using an LP solver on a constraint matrix that enforces the group fairness/balance constraints).  It also presents some experiments that demonstrate the method is more effective at finding high quality fair clusterings than existing deep clustering, fair clustering, and deep fair clustering methods.

**Summary Of The Review:**

The paper is generally clear and well written, and the empirical results are impressive.  Still, there are several opportunities to clarify the details of the approach as well as its advantages, to note its limitations, and to provide intuition for the empirical results (that may be surprising to some readers).

---

> ### Author Response · Authors · 2021-11-22
> **Response to Reviewer 5Bwe's main comments**
>
> We thank the reviewer for providing such a detailed review with so many helpful suggestions. We hope the following point-by-point response addresses your concerns.
>
> **Unfortunately, it is not very clear how this method is able to guarantee a fair clustering, ... I suppose the fairness guarantee involves using the output from the ILP in the last iteration as the output, but this doesn’t seem clearly specified.**
>
> The reviewer is correct that the output from the ILP in the last iteration is guaranteed to be fair. We denote these results as (Ours Results - ILP) in Table 1. For the learned deep model, optimizing our fairness objective along with the clustering objective cannot guarantee optimal fair results. However, we empirically demonstrate our deep model's performance to achieve a near-optimal solution with binary PSV. To our best knowledge, no existing deep fair clustering works guarantees fairness.
>
> **One limitation of this work seems to be that each person can only belong to one protected class, which doesn’t seem to be the case for (Bera et al., 19).**
>
> Thanks for pointing out this. (Bera et al., 19) have demonstrated their approach in a novel setting in which we have to balance the clustering results regarding multiple PSVs (e.g., gender and race of a person). Our approach can also be adapted to this setting by changing the total number of protected groups. For example, we have two groups for gender and four groups for race, to have an overall fair clustering we can have eight groups in total that consider each gender-race combination.
>
> **The importance of Theorem 3.2 (one of the stated contributions of the paper) is not very clear and could be emphasized.**
>
> Theorem 3.2 proves that optimizing our generalized fairness is equivalent to optimizing the general balance measure (Chierichetti et al., 2017) for disparate impact. In the approach section, we encode our fairness constraints based on the optimal condition of our generalized fairness definition. The experimental section shows that our proposed approach can achieve guaranteed balance results (this empirically demonstrates the correctness of Theorem 3.2), thus satisfying disparate impact.
>
> **For example, why are 2NK variables used (rather than NK), and doesn’t the number of variables typically equal the number of columns in the constraint matrix?**
>
> Yes, this is a typo, and the number of variables should be NK. We have a larger constraint matrix here because we encode all the different types of constraints into one constraint matrix as described at the end of section 4.2.
>
> **Why does beta=4 seem to be the sweet spot for all datasets and metrics? Any intuition for this (if any could be gained) would be interesting to report.**
>
> Note that we provided an unsupervised way to tune beta in section 5.3. This is a valuable insight to do model selection for deep fair clustering. When we observe that the model starts to converge to near-optimal balance, we can stop increasing the value of beta. It turns out to be a coincidence that beta=4 for both MNIST-USPS and HAR. We also tuned the beta for Reverse-MNIST, and the sweet spot is beta=6.
>
> **The authors note this and claim that fairness provides “positive guidance” for clustering performance, but this does not seem to be an explanation as much as a hypothesis. A third example, “we observe the ACC and NMI are decreasing with larger ε”, seems very counter intuitive (similar to the first example, but perhaps more so) and some explanation beyond “positive guidance” would be encouraging.**
>
> This does need further study. We observed the trade-off between fairness and clustering performance for traditional (tabular) fairness data sets in Figure 2 (c). Our approach achieved higher balance than k-means but worse clustering performance.   However, we note that all the existing deep fair clustering methods using two non-tabular data sets (MNIST-USPS and reverse-MNIST) have shown that fairness constraints can sometimes serve as "positive guidance" for clustering performance. Note that the ground truth for MNIST-USPS and reverse-MNIST achieve both optimal balance and ACC/NMI. This indicates that there need not be a trade-off between fairness and clustering performance in these two datasets but the conditions where this is the case needs further study.

---

> > ### Author Response · Authors · 2021-11-22
> > **Response to Reviewer 5Bwe's minor comments**
> >
> > Here we address the minor comments from the reviewer:
> >
> > **“Fairness takes two primary forms: i) group-level fairness and ii) individual-level fairness.” Please consider citing, I have seen other characterizations of fairness. “Different geographic regions place this tolerance at different levels.” Consider adding a citation.**
> >
> > Thanks for the suggestion. We have added the proper citations to those sentences.
> >
> > **“Most of these algorithms evaluate their performance on low-dimensional tabular data and mainly study the problems with binary PSV.” Most of and mainly seem like unfortunate hedges here. Perhaps it's worth pointing out which works evaluate their performance on low-dimensional tabular data and study the problems with binary PSV, and which do not.**
> >
> > We have updated the paper to include the works that use tabular data with binary PSVs.
> >
> > **“Finally, the base clustering model optimizes the clustering loss lC and lAug simultaneously” Doesn’t (Hu et al., 2017) optimize lC subject to constraints on lAug? (And isn’t this slightly different?)**
> >
> > Based on the implementation of (Hu et al., 2017) for clustering, they minimize the self-augmented loss and clustering loss together (equation 7 in their paper).
> >
> > **Let the rounded version of current assignment Y as Y’ ” I’m not sure I understand what is meant here (perhaps due to a typo resulting in a grammatical error).**
> >
> > Since $Y$ is a continuous variable, we round each soft prediction $Y_i$ into a hard prediction $Y^{'}_i$ which only contains binary values (e.g., [0.2, 0.5, 0.3] -> [0, 1, 0]). We have updated our paper to make it clear.
> >
> > **The choice of datasets makes sense (in that they were used in prior work). Nevertheless, using MNIST for fairness seems somewhat contrived and the results would seem more motivating on more reasonable data.**
> >
> > Following [1], we choose MNIST-related datasets to have a fair comparison. However, we agree that finding more reasonable datasets is better, which is one of our future directions.
> >
> > **What is ““fnlwgt”? It might be worth spelling out what the attributes mean/their possible values.**
> >
> > According to the Census dataset, "fnlwgt" means final weight. In other words, this is the number of people the Census Bureau believes the entry represents.
> >
> > **It seems like an overstatement to claim that by evaluating both accuracy and NMI the study is comprehensive (which means  including all or nearly all elements or aspects of something).**
> >
> > Thanks for pointing out this. We have removed the word comprehensively.
> >
> > **“Note we report both the deep model’s results and the final ILP’s results.” What is meant here does not seem to be specified anywhere and was confusing to me (and I think resulted in my confusion elsewhere). I think a sentence or two explanations could go a long way.**
> >
> > We are sorry for the confusion. The final ILP results are in the last row of Table 1, representing the ILP solver's results in the last iteration. The deep model's results mean our deep model's output after the last iteration.
> >
> > **“ground-truth and the guaranteed fairness results which are marked with blue”. I guess I know what is meant here, but there is a whole row labeled ground truth and only some of the cells in that row are blue.**
> >
> > Our goal is to mark the optimal fairness results as blue. We have removed the ground-truth part which causes your confusion.
> >
> > **If the train data came from a source where PSV information is available and the test data came from a source where it is not (e.g., the web, as described), is it reasonable to assume other changes in the population/resultant distribution are present as well?**
> >
> > Yes, the reviewer is correct. In Figure 2 (b), we observed that the population distribution changed for the HAR dataset so that the balance results are much lower than the optimal balance in the test set.
> >
> > **“We hypothesize this is because the fairness objective dominates the overall objective so that the impact of clustering objective is hindered” What is meant here? That training solely on the fairness objective does not lead to good clustering results? Something else?**
> >
> > Note that our fairness assignment problem is making the clustering assignments fairer by moving the instances near the clustering decision boundary (to minimize the total assignments changes). Training solely on the fairness objective is equivalent to making new assignments based on our ILP's initial output which has no guarantee on good clustering results. Through Figure 4, we empirically find out regularizing the fairness objective with the clustering objective yields better clustering results. One hypothesis could be that the clustering objective can prevent the fairness objective from moving the points far away from the clustering decision boundary.
> >
> > [1] Li, Peizhao, Han Zhao, and Hongfu Liu. "Deep fair clustering for visual learning." Proceedings of the IEEE/CVF Conference on Computer Vision and Pattern Recognition. 2020.

---

### Decision · Program_Chairs · 2022-01-20

**Decision:**

Reject

**Comment:**

This paper received a majority voting of rejection. In the internal discussion, no reviewer would like to change the score according to the author response. I have read all the materials of this paper including manuscript, appendix, comments and response. Based on collected information from all reviewers and my personal judgement, I can make the recommendation on this paper, *rejection*. Here are the comments that I summarized, which include my opinion and evidence.

**Motivation**

The motivation of this paper is not strong. In this paper, the authors claimed that the fairness level of deep clustering methods is relatively poorly compared with the traditional fair clustering methods. The traditional fair clustering methods employ the hard constraints to achieve fairness by scarifying the cluster utility. Instead, deep fair clustering methods seek the trade-off balance between fairness level and cluster utility; therefore, the deep fair clustering can be regarded to use the soft constraints. There is no necessary to compare two different fairness constraints. Even the proposed method is a trade-off balance between fairness level and cluster utility.

**Self-augmented Training**

The relationship between self-augmented learning and fairness learning is unclear. I guess that the authors added this modular simply to enhance the cluster utility. However, such a loss or an operator can be also applied to other (fair) clustering algorithms. The experimental comparisons in Section 5 is unfair. No ablation study on this is provided.

**Novelty**

One reviewer pointed out there existed some work that plugs integer linear programming into a probabilistic discriminative clustering model proposed in 2017.

**Experiments**

(1) ScFC and DFCV release their codes; no results of these two methods were reported on HAR. (2) No standard deviation. (3) The Initial ILP Results (Ours) and Ours Result in Table 1 on HAR dataset is 0.653 and 0.468, both higher than the Ground Truth (Optimal) 0.458.

**Presentation**

 A few statements are not well-supported, or require small changes to be made correct.

No objection from reviewers was raised to again this recommendation.